# A Contemporary Review of Aluminium MMC Developed through Stir-Casting Route

**DOI:** 10.3390/ma14216386

**Published:** 2021-10-25

**Authors:** Poonam Yadav, Alok Ranjan, Harish Kumar, Abhishek Mishra, Jonghun Yoon

**Affiliations:** 1BK21FOUR ERICA-ACE Center, Department of Mechanical Engineering, Hanyang University, Ansan-si 15588, Korea; poonamtusha@gmail.com; 2Department of Mechanical Engineering, National Institute of Technology Delhi, Delhi 110040, India; alokranjan@nitdelhi.ac.in (A.R.); harishkumar@nitdelhi.ac.in (H.K.); Abhishekmishra@nitdelhi.ac.in (A.M.)

**Keywords:** aluminium alloys, reinforcements, composites, microstructure, mechanical, wear properties

## Abstract

The growing demand for composite materials with improved properties is attracting a lot of attention from industries such as automotive, aerospace, military, aviation, and other manufacturing. Aluminium metal matrix composites (AMMCs), with various reinforcements such as continuous/discontinuous fibers, whiskers, and particulates, have captured the attention due to their superior tribological, mechanical, and microstructural characteristics as compared to bare Al alloy. AMMCs have undergone extensive research and development with different reinforcements in order to obtain the materials with the desired characteristics. In this paper, we present a review on AMMCs produced through stir casting routes. This review focuses on the following aspects: (i) different reinforcing materials in AMMCs; (ii) microstructural study of reinforced metal matrix composites (MMCs) through stir casting. Both reinforcing micro- and nanoparticles are focused. Micro- and nanoreinforced AMMCs have the attractive properties of combination such asthe low-weight-to-high-strength rati and, low density; (iii) various tribological and mechanical properties with the consideration of different input parameters; (iv) outlook and perspective.

## 1. Introduction

The combination of properties (such as high modulus and strength, high abrasive wear resistance, high toughness, and low deformation) are highly desirable in the areas of aviation, electronic devices, automobiles, space shuttles, and marine industries [1,2]. However, monolithic material’s inability to have this attractive properties combination make metal matrix composites (MMCs) as a suitable successor for monolithic materials, as these desired properties combination can be induced in MMCs [1,3]. Extensive studies carried out in MMC areas reveal that either MMCs are in their developmental stage or their manufacturing processes are still not well-established. In addition, the high fabrication cost of MMCs is the biggest hurdle in their applications [4,5]. Generally, the fabrication of MMCs demands lower-weight and -density components. Currently, popular matrix materials are aluminium, magnesium, and titanium.

Various nano- and microparticulates reinforcements, such as silicon nitride, boron, magnesium oxide, mica, boron nitride, titanium carbide, silicon carbide (SiC), glass beads, alumina, boron carbide titanium boride, and silica sand, are used by researchers [6]. Reinforcing material greatly influences the composite properties. For example, zirconium imparts high resistance to abrupt change in volume at elevated temperature and high refractoriness to the composites [7,8]. Various techniques to fabricate reinforced aluminium metal matrix composites (AMMCs) along with their microstructure and mechanical properties are reported [9]. Since a limited number of review articles are available in the area of reinforced AMMCs including fabrication methods, microstructures, and mechanical and tribological properties such as tensile, density, hardness, and wear [10]. Kala et al., Surappa et al., and Rajmohan et al. reviewed the mechanical and tribologicalcharacteristics of stir-cast composites [11,12,13]. Ramnath et al. and Sharma et al. presented the review on the merits and demerits of different reinforcements in aluminium alloy MMCs [14,15]. Ramanathan et al. presented a review on the significant aspects of the stir casting process, which includes composite characteristics, furnace design, obstacles in composite production, and possible research prospects [16]. Sharma et al. studied a literature review on the fabrications of AMMCs through stir casting routes [17]. Many researchers are focused on Al6061 MMC produced by stir casting routes. Kareem et al. presented a review on AA6061 MMCs produced via stir casting routes [18]. Few researchers focused on nano and hybrid AMMCs. Muley et al. provided an overview of various developments in Al-based nano and hybrid composites [19]. Khalid et al. reviewed on the effect of processing parameters such as the stirrer shape and its dimension, stirrer velocity, stirring temperature, stirring duration, and preheat temperature to which the reinforcement and the mold are subjected, reinforcement feed rate, wetting agent, and melt pouring, as well as challenges encountered in successfully fabricating AMMCs via the stir casting technique [20]. Still there is a need for a detailed review of recent encompassed areas for reinforced AMMCs. Our focus is to review on different reinforcing materials in AMMCs and study their microstructural, mechanical and tribological properties through stir casting routes.

## 2. AMMCs with Various Reinforcing Materials

Some of the most often used AMMCs are Al reinforced with SiC, Al_2_O_3_, or B_4_C, which produce superior mechanical qualities at a cheaper production cost. As a result, many engineers have used AMMCs for a variety of applications, including brake rotors, drive shafts, pistons, and cylinder liners. Reinforcing materials are classified as industrial wastes, agricultural waste derivatives, and artificial ceramics particulates. The ultimate properties of AMMCs are determined by the matrix alloy and the specified reinforcement. The characteristics of the reinforcements and the matrix alloy are used to choose the processing procedure for making AMMCs.

### 2.1. AMMCs Reinforced with SiC

Researchers have observed that addition of SiC as a reinforced particle with AMMCs yields superior machinability and mechanical properties. Ozben et al. found that a higher reinforcement ratio enhances the hardness, tensile strength, and density but reduces the impact toughness [21]. Ozden et al. analyzed the impact behavior of a SiC-reinforced aluminium alloy [22]. AMMCs reinforced with SiC and Al were investigated under different temperatures by the authors to know the impact behavior. The weak matrix–reinforcement bonding, the agglomeration of particles, and the particle clustering affect the impact behavior. However, the test temperature effect on the material impact behavior is least. With increase in the temperature, the strength, ductility, and modulus of the composite are decreased. Srivatsan et al. identified that an under-aged microstructure’s cyclic fatigue life deteriorates more as compared to that of a peak-aged microstructure [23]. In addition, a higher load ratio results in an enhanced fatigue strength for a given ageing condition. Thunemann et al. [24] investigated the properties of preceramic-polymer-bonded SiC AMMCs. The binding agent was polymethylsiloxane (PMS). The result showed that 1.25 wt% of the polymer brings stability to enable the processing of composite. It was found that the PMS-derived binder provides a desired strength to SiC without degrading its mechanical characteristics and properties. Sujan et al. studied the physio-mechanical characteristics of Al_2_O_3_ and SiC-incorporatedAMMCs [25]. Few researchers have investigated SiC and Al_2_O_3_-reinforced MMCs. The investigation confers the enhancement in mechanical as well as physical properties such as increased ultimate tensile strengths (UTSs) up to 23.68%, a low thermal expansion coefficient (4.6 × 10^−6^/°C), andincreased strength-to-weight ratios. The authors found that the rockwell hardness for 100% Al is 25 but when 15% SiC is added its hardness increases upto 50. As compared to the Al–Al_2_O_3_ composite, the Al–SiC-reinforced composite was found to have a lower wear rate. The particle clustering and its impact on the flow behavior of SiC-reinforced AMMCs were examined by Zeng et al. [26]. It was found that clustering has a bigger impact on the mechanical behavior of the matrix during tensile deformation than on the elastic response and that plastic deformation is also significantly affected by particle clustering. Compared to a particle random distribution, the microstructure with particle clustering is subjected to a higher particle fracture percentage. Prabhuet et al. investigated the impact of the stirring duration andthe speed on particle dispersion in SiC-reinforced AMMCs [27]. The study looked at how different stirring speeds and periods can be used to manufacture a high-silicon-content aluminium alloy with 10% SiC. More particle clustering in some locations occur at a lower stirring time and speed, and a better distribution occurs at higher values of the stirring speed and duration. At 600 rpm with 10 min stirring resulted in uniform hardness. Barekar et al. obtained regular stir casting results of particles agglomeration that induces very low ductility in composites [28]. Palanikumar and Karthikeyan as well as lKylyckap et al. optimized the parameter such as the feed rate, the SiC% volume fraction, and the cutting speed to obtain the minimum surface roughness [29,30]. For surface roughness, feed is the most influencing parameter, followed by the vol% fraction and the cutting speed of SiC. For rough turning and medium turning, the high feed and the high depth of cut with a low cutting speed are recommended. To obtain a good surface finish with a composite-coated carbide-cutting tool, a lower feed rate with a higher cutting speed is recommended. Natrajan et al. compared the wear behavior of the A356/25SiC MMC against an automobile friction material with that of grey cast iron reveals and reported that the composite has a high wear resistance, which makes it a suitable candidate for drum brake lining materials [31]. However, due to the presence of hard silicon carbide particles (SiCps), it cannot be used as a lining material. Yanming and Zehua examined the tool wear behavior during the machining of a SiCp-reinforced composite depicts abrasive wear as a major damaging mechanism on the tool flank edge for conventional tool, whereas for a high hardness tool brittle failure is the major damaging mechanism [32]. The size of SiCps and the SiC volume percentage are vital parameters affecting the tool life.

### 2.2. AMMCs Reinforced with Aluminium Oxide (Al_2_O_3_)

Park et al. analyzed the impact ofthe incorporation of Al_2_O_3_ in aluminum for weight percents ranging from 5% to 30% and discovered that the weight percentage of Al_2_O_3_ affects the MMC’s fracture toughness [33]. With the increasing volume fraction of Al_2_O_3_, the fracture toughness decreases due to interparticlegaping among nucleated microvoids. Park et al. examined the fatigue characteristics of Al6061–Mg–Si alloy-reinforced alumina (Al_2_O_3_) microspheres with volume fractionsvarying from 5% to 30% [34]. The resultsrevealed that the liquid metallurgy and unreinforced alloy-processed composite has a lower fatigue strength compared to powder metallurgy (PM) composites. Tjong et al. reported a comparison between the characteristics of aluminium composites, namely Al–B–TiO_2_ system and Al–B_2_O_3_–TiO_2_ system [35]. It was noticed that ceramic formation (TiB_2_ and Al_2_O_3_) resulting from the composite hot pressing together with the formation of Al_3_Ti such as a lower Al_3_Ti content results in a higher fatigue strength and a higher Al_3_Ti content results in high tensile strength. Kok used the vortex method to fabricate an Al_2_O_3_-particle-reinforced 2024 Al alloy composite and studied its mechanical properties [36]. The optimum production process condition is preheating the mold at a temperature of 550 °C, a stirring speed of 900 rev/min, a stirring time of 20 min, a pressure of 6MPa, a pouring temperature of 700 °C, and a particle incorporation rate of 5 g/min. Pressure improves the bonding and wettability between aluminium alloy/alumina particles but decreases the porosity. Kumar et al. carried out the characterization of anA359/Al_2_O_3_ MMC produced by electromagnetic stirring casting and revealed that the increase in the tensile strength of the matrix composite and stirring (electromagnetic-assisted) results in a reduced grain size and enhances the bonding at the particulate–matrix interface [37]. The investigations of the wear resistance and the hot deformation of AMMCs produced via PM reveals that the Al_4_C_3_ and Al_2_O_3_ addition improves the hardness and the compressive strength (CS) [38,39]. In addition to that, Al_4_C_3_ also improves the mechanical properties of MMCs. The microhardness variation in machined Al_2_O_3_-reinforced AMMCs with particulate properties and cutting parameters was studied. In the neighborhood of the machined surface layer, a higher microhardness was found. A low volume percentage reinforced with coarse particles results in a higher micro-hardness variation.

### 2.3. AMMCs Reinforced with Boron Carbide (B_4_C)

Yao et al. [40] reported that the strength of trimodal aluminium metal–matrix composites is influenced by microstructural characteristics. The authors studied the factors affecting the trimodal AMMCs strength and reported that high strength is obtained with amorphous and crystalline aluminium nitride and Al_4_C_3_, very high dislocation densities in the NC–Al and CG–Al domains, the nanoscale dispersion of Al_2_O_3_ interfaces between the different constituents, and the distribution and concentration of nitrogen. Vogt et al. [41] examined the cryo-milled aluminium and B_4_C nanocomposite plates made by the following route:(1) three-step quasi-isostatic forging (QIF); (2) hot isostatic pressing (HIP) accompanied by two-step QIF; (3) HIP accompanied by high-strain-rate forging (HSRF). The results showed that QIF plates have higher ductility but a lower strength compared to HIP/HSRF plates. HSRF causes inhibition to dynamic recrystallization, resulting in the enhanced strength and the decreased ductility of HIP/HSRF plates. Babu et al. [42] carried out aninvestigation of surface quality during the machining of hybrid composites (Al–SiC–B_4_C). The researchers used the taguchi method to study surface roughness characteristics of the machined hybrid Al–SiC–B_4_C MMC. The results showed that feed is the dominant parameter accompanied by the cutting velocity. However, the feed rate effect on the surface quality is insignificant. Previtali et al. used the traditional investment casting method to fabricate aluminium matrix composites [43]. The comparison of SiC and B_4_C as a reinforcement to aluminium alloy reveals the higher wear resisting capability of SiC-reinforced MMCs compared to B_4_C-reinforced MMCs.

### 2.4. AMMCs Reinforced with Fibers

Sayman et al. analyzed the in-plane elasto-plastic stress behavior of AMMC-laminated plates [44]. They reported an increment in the load bearing capacity of laminated plate with a good matrix–fiber bond observed at a pressure of 30 MPa and at a temperature of 600 °C. Sayman et al. studied the effect of AMMC beams, which are subjected to the elastic–plastic thermal stress [45]. In that study, the investigation of a steel-fiber-reinforced AMMC beam under elastic–plastic thermal stress revealed that the greatest equivalent plastic strain and residual stress occur at an orientation angle of 0°. In addition, for a higher orientation angle, plastic yielding occurs at a lower temperature. Atas and Sayman examined the expansion of the plastic zone in aluminium metal–matrix-laminated plates using elastic–plastic stress analysis [46]. In laminated plates of steel-fiber-reinforced the AMMCs yielding starts from edges; however, at the corner of plates, no yielding takes place. Ding et al. investigated the short-fiber-reinforced aluminium matrix composites with low-cycle fatigue crack development and life prediction [47]. Under controlled strain conditions, a low-cycle fatigue test was carried out in 20 wt% Al_2_O_3_-reinforced Al. Within a wide temperature and a strain amplitude ranges, the cast fatigue lives match with the noticed fatigue live. The best matches between the predicted fatigue lives and the observed fatigue lives occur at a larger cyclic plastic strain and a larger total strain. Ding et al. investigated ashort-fiber-reinforced 6061 AMMC with a low-cycle fatigue model. The Al_2_O_3_ incorporation in the Al6061 matrix results in a microstructural strengthening and a fatigue ductility reduction [48]. Lee et al. discovered the attributes of a carbon-fiber-reinforced 7075 AMMC as a function of temperature and strain rate [49]. A high temperature lowers the flow stress, while a high strain rate results in a higher flow stress. The increases in strain and temperature result in lowering the work hardening rate. Gudena and Hall studied a continuous-fiber-reinforced AMMC’s deformation characteristics at high strain rates [50]. They investigated the compressive deformation response of an Al_2_O_3_-reinforced AMMC at larger strain rates in the transverse as well as longitudinal directions, and it was established that the composite strain rate in the transverse direction is similar to for monolithic alloy. Rams et al. studied aluminium matrix composites that were reinforced with electroless nickel coated with short carbon fibers [51]. It was reported the composite wettability increases with electroless nickel coated with fibers as a reinforcement. This is attributed to the formation of Ni–Al–P transient intermetallic layer because of heating. Fu et al. studied the wear characteristics of Saffil/Al, saffil/Al_2_O_3_/Al, and saffil/SiC/Al hybrid MMCs [52]. The best wear resistance of the saffil/Al composite is obtained. At a normal temperature, compared to that of the saffil/Al/SiC composite, the wear resistance of the saffil/Al_2_O_3_/Al composite is better, while the wear resistance of the saffil/Al/SiC composites better than that of the saffil/Al_2_O_3_/Al composite at a higher temperature.

### 2.5. AMMCs Reinforced with Zirconium

Zirconia (ZrO_2_) is one of the most cost-effective and widely available, with the ability to maintain high strength at high temperatures and outstanding mechanical and wear qualities. Zirconia has excellent qualities such as high strength, hardness, and wear resistance. ZrO_2_ particles in aluminium alloys have significantly improved mechanical and wear properties [53]. Jeno et al. evaluated the characteristics of an Al6063 matrix reinforced with zircon sand and alumina (Al_2_O_3_) with volume fractions combinations of zircon sand and alumina of 0% + 8%, 2% + 6%, 4% + 4%, 6% + 2%, and 8% + 0% [54]. The composites’ hardness values and tensile strengths are higher for the volume fraction combination of zircon sand and alumina (i.e., 4% + 4%). In this combination, the uniform particle distribution occurs with fewer pores sites. Das et al. [55] investigated the effect of alumina and zircon sand as the reinforcement. They reported that the incorporation of zircon sand and alumina enhances the wear resisting property of the Al–Cu alloy. Compared to the alumina-reinforced composite, a better particle matrix bonding occurs in the zircon-reinforced composite, resulting in an enhanced wear resistance. Scudino et al. [56] examined the mechanical behavior and characterization of PM-produced Al-based MMCs augmented with Zr-based glassy particles. It was found that the addition of 40% zircon-based glassy particles in the Al-based matrix increases the CS by 30% while a 25% increment in CSwas reported with the incorporation of 60% zircon-based glassy particles.

### 2.6. AMMCs Reinforced with Fly Ash (FA)

FA particles (SiO_2_, Al_2_O_3_, and Fe_2_O_3_ are FA constituents) are thermal power plant wastes having a low density and a low cost, which act as potential reinforcements. Rajan et al. examined Al–7Si–0.35 Mg/FAMMCs made by various stir casting methods [57]. They reported that the modified compocasting cum squeeze casting enhances the CS of the composite compared to that of the matrix alloy but it reduces the tensile strength. The modified compocasting cum squeeze casting results in a better distribution of the porosity-free FA composite. Dou et al. studied the effectiveness of an aluminium alloy–FA composite in terms of electromagnetic shielding [58]. The tensile strength of the composite decreases due to the impartation of electromagnetic shielding characteristics to the composite. Ramachandra and Radhakrishna [59] studied the corrosive and sliding wear behavior of aluminium matrix composites with an FA reinforcement. The researcher experimentally found that the incorporation of FA results in a wear resistance increment and a corrosion resistance decrement. With the decrease of the sliding velocity and the normal load, the wear resistance decreases.

## 3. AMMCs Produced through Various Processing Techniques

### 3.1. Processing of AMMCs

Ex situ and in situ syntheses are the methods for fabricating MMCs. In ex situ synthesis reinforcements are added from outside, whereas in in-situ synthesis the reinforcement is achieved by chemical reaction, e.g., exothermic dispersion and direct melt reaction [4,5,16,19]. Ex situ methods consist of solid-state processing and liquid-state processing. MMCs can be fabricated through many methods, but the most popular methods are liquid-state and solid-state fabrication methods. Liquid-state fabrication methods for MMCs are fairly simple and less costly as compared with solid-state ones [17]. Table 1 shows some fabrication techniques with unique features and costs [60,61].

The flow chart for the production process for MMCs is shown in Figure 1. As indicated in Figure 1, the MMC production procedures can be characterized as either primary operations (liquid or solid metal processing) or secondary processes (semi-solid, in situ, and others). AMMCs could be customized according to the area of application by choosing a proper reinforcement for a given matrix, parameters, and additives used to improve the AMMCs quality with the stir casting method. Many researchers studied the effects of process parameters on different matrices and reinforcement materials [16,27,62,63,64,65,66,67,68]. Ramanatham et al. discussed the important process parameters and additives of the stir squeeze casting method, which is shown in Figure 2 [16]. The squeeze pressure, reinforcement size, stirring velocity, stirring duration, stirrer blade design, and die preheating temperature are some of the factors. The squeeze pressure is the most influential characteristic out of all of them. The majority of previous studies concluded that a squeezing pressure of 100 MPa is sufficient for grain refinement and reduced porosity. There are a variety of furnace configurations that could be utilized to make AMMCs. A bottom pouring electromagnetic-assisted stir casting set-up with an option of ultrasonic stirring especially for nanomaterials, as shown schematically in Figure 3, is recommended for the fabrication of MMCs [16].

The agglomeration and clustering tendency of the possessed nanoparticles isseen because of their high surface energy, attractive Vander Waals force, and electrostatics which alters their homogenous distribution while processing. Many researchers used the ultrasonic process to manufacture nanoparticle composites and noted the uniform spreading of nanoparticles which occurddue to the inhibition of particles clustering [69,70,71,72,73,74,75,76,77]. Zok et al. [78] examined the thermo-mechanical characteristics of ahybrid composite that was fabricated using the extrusion technique for amalgamating an aluminium/SiC/aluminium composite.

Jang et al. fabricated agraphite-reinforced MMC by PM and investigated its physical and mechanical behaviors [79]. The investigators noticed that a nanomaterials surface properties alteration and a high aspect ratio (AR = 1000) result in individual fiber clustering into agglomeration. The ultrasonic dissemination of the nanosized fiber material results from the low-energy sonication application. Mula et al. investigated an ultrasonically cast Al-2%Al_2_O_3_ nanocomposite structure and studied itsmechanical qualities [80]. It was found that the mechanical properties achieved by the uniform distribution of alumina nanoparticles in aluminium resulted from the use of noncontact ultrasonic casting for nanocomposites (Al/Al_2_O_3_). Lin et al. investigated the microstructure of a nanostructured AA2924–SiCMMC that was fabricated usingthe milling and hot pressing technique [81]. It was found that at elevated temperatures, recrystallization and grain growth occur, which results in bimodal grain size scattering and a nonhomogenous distribution. Liu et al. studied the amalgamation of a SiC/Alcomposite under high pressure and found that a low volume-to-surface area ratio of nanoscale reinforcement makes the diffusion and reaction happen easily, resulting in a better matrix reinforcement bonding [82]. However, the complete agglomeration is unavoidable. Torralba et al. used the PM to fabricate aluminium matrix composites and noticed no undesired and interfacial reaction, which can be attributed to low temperature that is involved in PM [83]. With the PM technique, SiC/Ti alloy MMCs manufacturing is possible which cannot be manufactured by other methods.

Woo and Zhang used PM and ball milling to produce an Al–7 wt% Si–0.04 wt% Mg/SiC-reinforced composite [84]. It was found that during machining eutectic silicon formation happens which the rate of compacting is boosted due to an enhanced composite powder diffusion rate, resulting in a fine microstructure formation with less pores and a higher hardness value.

Zhang et al., Sureshbabu et al., Feng et al., and Geng et al. used the squeeze casting technique to manufacture advanced composites which distribute particles in nanoforms homogeneously that improves fracture toughness [85,86,87,88]. Reddy et al. investigated an aluminium hybrid nanocomposite using mechanical activation and solid-state combustion [89]. For enhancing bulk properties, in situ manufactured composites are superior to ex situ manufactured composites.

Reddy et al. [89] investigated ahybrid aluminium nanocomposite using mechanical activation and solid-state combustion. For enhancing bulk properties, in situ manufactured composites are superior to ex situ manufactured composites. Wang et al. manufactured an aluminium-matrix composite through the in situ method (direct melt reaction) with Al_2_O_3_p as a reinforcement material [90]. The observations indicate that there is a homogenous distribution of Al_2_O_3_ particles and a clear interface between the particles and the Al metal matrix. Huge fine sub-grains near alumina particles, high-density dislocation, and no impurity existence between the matrix and the reinforcement result in the enhanced performance of the composite. In addition, the in situ method imparts isotropicity to the fabricated composite. Muley et al. studied hybrid nanocompositesand reported that due to the difference of the densities between the matrix and the reinforcement, particles settling occurs during the liquid-state processing of the aluminium based hybrid nanocomposite, which can be avoided by stir casting method [19]. Rajan et al., Ramesh et al. and Schultz et al. investigated the conventional stirring that isaccompanied with casting and reported that this is the most economical process for metal matrix production [57,91,92]. During stirring, reinforcement nanoparticles are not wetted, and also nanoparticles tend to cluster. This problem can be overcome by high shear stirring combined with reactive wetting. Hashmi et al. studied the problems encountered in the fabrication of AMMCs by stir casting, such as wettability, uniform distribution, and low porosity [93]. The stir casting method can manage large production quantity, decreases overall manufacturing cost and allows big-size components fabrication. Hot extrusion is a forming method and is mostly used to enhance AMMCs mechanical properties. Sharifian et al. examined the tensile characteristics of hot extruded Al-based composites with various Al_4_Sr contents and reported that hot extrusion reduces the porosity, resulting in the enhancement of percent elongation and the ultimate tensile strength of AMMCs [94]. The optimum values of UTS and elongation are obtained with extrusion at 420 °C with 18:1 as an extrusion ratio. Alizadeh et al. analyzed the impact of the hot extrusion method on the physical and mechanical behaviors of an Al-based composite [95]. It was found that extrusion facilitates the homogeneous dissemination of the reinforcing particles in the aluminium matrix, imparts a fine size to the grains, agglomerates crack and enhances the matrix particle bonding. The porosity, reinforcement dissemination, and microstructure influence the ductility of AMMCs. Muley et al. studied various manufacturing techniques and reported that the most appropriate production technique should be selected to produce Al-based hybrid composites and nano-composites [19]. The selected production technique can result in a uniform distribution of the reinforcement without agglomeration and clustering, enhances the matrix-reinforcement bonding, improves wettability, refines grains, generates a clear and nonreactive interface, lessens grain growth, lowers the processing temperature, optimizes factors capable of processing a huge amount of reinforcement, improves mechanical properties, lowers the thermal expansion coefficient, and lowers the production cost.

### 3.2. AMMCs through Stir Casting Routes

There are different kinds of aluminium alloys which have their own advantages and applications such as Al6061, A2024, and A7075. The enhancements in mechanical and tribological properties are achieved by the incorporation of alloying elements, such as Si, Mn, Mg, and Cu, into AMMCs [96,97,98]. Rashmi et al. [99] found that A2024 which reduces the weldability and corrosion resistance, but it is suitable in the automobile sector for light-weight appliances. The Mg and Si presence makes Al6061 most versatile with the strength between those of A2024 and A7075. A7075 alloy has seaside applications and is prepared with zinc and magnesium as alloying elements which impart high strength and corrosion resistance. LM6 and LM25 alloys can be reused for the preparation of MMCs because of its recyclability without purification processes [100,101]. Al 7075is the best material for high-strength applications, since its UTS can range from 280 to 570 MPa depending on the heat treatment condition. The addition of a reinforcement improves the MMC’s strength even more. Both LM6 (Al–Si_12_Al–Si_12_Fe) and LM25 (Al–Si_7_Mg) have good fluidity, making complex shapes easy to cast. They also have great corrosion resistances. A wide variety of reinforcements have been used to fabricate AMMCsto enhance their mechanical and tribological properties [17,18]. Khanna studied the mechanical properties of an AMMC by adding graphene/carbon nanotubes, which enhances its advancement and opportunities in industries [102]. Table 2 represents various AMMCs with different reinforcements fabricated via stir processing routes.

## 4. Properties of AMMCs through Stir Casting Routes

### 4.1. Microstructural Studiesof Reinforced AMMCs

Compared to unreinforced Al/Al alloy, AMMCs have superior mechanical properties (high stiffness, low density, high strength, and enhanced elevated properties) because of altered matrix microstructures [12,113]. AMMCs characteristics, such as thermo-mechanical, physical, tribological, and mechanical properties, are highly dependent on reinforcements including zirconium, carbon nanotubes (CNTs), FA, Al_2_O_3_, and SiC. The uniform dispersion of reinforced particles in the matrix, better matrix–reinforcement bonding, and grain size refinement are the major causes for the improvement in the mechanical properties. A finer structural morphology provides better mechanical properties than a coarser morphology. In the las few decades, hard ceramic particles have been used to reinforce AMMCs composites. Banerji et al. proposed the process of preparing pure Al and 11.8% Si–Al composites containing zircon particles manufactured through the stir casting method [114]. In this work, it was discovered that the aforesaid two melts needed to be alloyed with 3% Mg (25 to 30%. Furthermore, by adding up to 5% Mg, it was possible to distribute up to 60 wt% zircon. The separation of magnesium and silicon was detected at the matrix–particle grain boundaries, and zircon was certified by EPMA investigation. The alloy with Si (11.8%) and composites carrying 10 wt% zircon and 30 wt% zircon displayed substantial eutectic silicon structure refinement because of heterogeneous nucleation sites formation by diffused zircon particles. The size of the proeutectic Al decreases with the increments in zircon percentage and eutectic silicon nucleation was noticed adjacent to zircon sand. The pressure-die-cast MMCs carrying 60 wt% zircon sand with sizes of 40–100 µm have a uniform particle diffusion within the cast Al-alloy metal matrix. Panwar et al. Analyzed the wear debris and characteristics of a stir-cast LM13/Zr composite at high temperatures and reported thatthe zircon reinforcement in the Al–12% Si alloy MMC results in an unvarying particles distribution, showing a better matrix–reinforcement bonding [115]. Das et al. and Sharma et al. analyzed the age hardening of stir-cast zircon sand/Al–4.5 wt% Cu composites [116,117]. In the aluminium–4.5 wt% copper alloy, the addition of SiCp and zircon sand results in a secondary phase creation (Al_2_Cu) at the interdendritic region.

It was found that aluminium–4.5 wt% copper composites possess a cellular structure along with unvarying zircon sand (ZrSiO_4_) throughout the metal matrix (Figure 4a) and a secondary phase rich in copper (CuAl_2_) at the particle–matrix interface (Figure 4b).

Das et al. [55] critically analyzed the wear behavior of an alumina-reinforced Al–4.5 wt% Cu composite. It has been shown in (Figure 5a,b) that zircon, alumina, graphite, and silicon are utilized as reinforcements to develop aluminium alloy hybrid composites, which improves the abrasion resistance, the wear resistance, and the hardness of the composites. Compared to alumina particles, zircon induces a uniform particles distribution tendency and promotes bonding. Therefore, for developing the Al–4.5 wt% Cu alloy’s microstructure, zircon addition is preferred over alumina particles.

Gopi et al. carried out the characterization of aluminium 6061/zircon sand/graphite particle hybrid composites [118]. It was reported that primary aluminium dendrite boundaries with coarse acicular intermetallic particles are present in zircon and graphite-reinforced Al6061 alloy hybrid composites. The researchers reported that the heat treatment of the composite of Al6061 alloy reinforced with zircon sand and graphite particles results in a refined grained structure, which is in accordance with other researches.

Li et al. examined the mechanical and ageing behaviors of zircon-reinforced Zn–4Al–3Cu alloy [119]. The researchers discussed the microstructure of ZAS alloy. Figure 6 shows a hypoeutectic dendritic phase having Zn-rich primary-phase dendrites.

Kumar et al. and Sharma et al. investigated a dual-reinforced-particle (DRP) aluminium alloy composite [120,121]. A fine distribution of globular eutectic silicon near the reinforced particles occurs with zircon and SiC as the reinforcements. Kumar et al. examined the wear characteristics of azircon-sand-reinforced aluminium alloy [122]. The reinforcement of LM13 alloy with zircon particles results in a uniform reinforcement particles distribution with porosity and particle clustering.

Abdizdeh et al. studied an ex situ ZrSiO_4_-reinforced MMC [123]. Dendritic arm silicon present with the coarse size of dendrites is shown. The SEM images of the composite (Figure 7a,b) show the change in structure of eutectic silicon to dense globular from acicular in the surrounding area of zircon sand, together with a grain particle homogenous distribution.

Kalaiselvan et al. [124] examined the fabrication and characterization of an AA6061 MMC reinforced with B_4_C particulates. In this study, B_4_C presence in the matrix was detected from XRD analysis shown in Figure 8.

With the B_4_C content increment, the peak of Al decreases, while the peak of B_4_C increases. Compared to that of the base alloy, the Al peak is deviated towards lower. The XRD analysis confirms there is no reaction between B_4_C particles with the aluminium matrix, which can be attributed to the reaction barrier created due to the formed layer of Ti around B_4_C, preventing the interfacial reaction between aluminium and B_4_C.

Satheesh and Pugazhvadivu reinforced a SiC–coconut shell ash (CSA) composite with the stir casting process [125]. The SEM micrographs of the cut sectional surfaces of the cast Al6061 and its composites are shown in Figure 9a–g. It can be seen in Figure 9b–f that the dispersion of the reinforcement particles is uniform, with no agglomeration.

However, the SEM micrograph of the Al6061–SiC–10% CSA composite (Figure 9g) revealed agglomerations of the reinforcement and voids. It can be inferred that adding up to 8% CSA to the matrix results in a good reinforcement dispersion. From the XRD result shown in Figure 10, the authors found that Al predominant peaks are at 38.4°, 44.71°, 65.1°, and 78.2° (JCPDS No. 89-4037) and this indicates a good dispersion. No visibility of minor peak confirms that no impurities are present in Al6061 hybrid composites. As wt% of Al_2_O_3_ and Fe_2_O_3_ are less than 5%, other peaks of the hybrid composite are not visible.

### 4.2. Mechanical Studiesof Reinforced AMMCs

Many researchers found that the B_4_C addition increases the mechanical property of Al6061. Gudipudi et al. demonstrated that ultrasonication-assisted stir casting is a viable option for achieving microstructural changes that improve characteristics [126]. Individual B_4_C distribution and microstructure refinement were accomplished at 4 wt% B_4_C in this investigation. At a 4 wt% B_4_C, the specific ultimate and the CS increased by 36.32% and 43.92%, respectively, while specific Vickers and Brinell hardness values increased by 53.41% and 50.89%, respectively. The mechanical properties of matrix alloy AA6061with the incorporation of B_4_C was studied by Kalaiselvan et al. [124]. They observed that as the reinforcement particulates increases, the micro- and macrhardness values of AMMCs are linearly increasing. As the volume fraction of reinforcing particulates increases, the reinforcement increases resulted in the reduction of thematrix grain size. Sam et al. studied the effects of carbide ceramics (B_4_C, SiC, and TiC) as reinforcements on the recip-rocating tribology performance and mechanical strength of A333 hybrid composites, which were compared to those of the alloy [127]. In this study, the A333/B_4_C/SiC hybrid composite shows a dominating trend compared to the A333/B_4_C/TiC composite, and the alloy displays the lowest average hardness. As the reinforcement surface area increases, plastic deformation tendency increases, resulting in hardness increases. Ramesh et al. studied the mechanical properties of Ni–P-coated Si_3_N_4_-reinforced Al6061 composites and also discusses the fabrication of Ni–P-coated silicon-nitride-reinforced Al6061 composites by the stir cast method [91]. The addition of Ni–P-coated silicon nitride particles to the matrix alloy improves microhardness significantly. This increase in the matrix alloy hardness can be due to the fact that silicon nitride, as a hard reinforcement, exhibits a better resistance to hardness tester indentation by imparting its inherent hardness to the matrix alloy. The presence of the hard ceramic phase induces hardness, as a reduction in ductile metal content into composites occurs. Satheesh and Pugazhvadivu examined the mechanical properties of Al6061 and its composites (Al6061–SiC/CSA hybrid composites) [125]. Figure 11 shows the densities of Al6061 and its composites. Due to the higher density of SiC particles (3.10 g/cm^3^) compared to that of Al6061 (2.7g/cm^3^), the density of the Al6061–SiC composite is higher. Further, it can be noted that, as the density of CSA particles (1.65 g/cm^3^) is lower than those of Al6061 and SiC, the increase in CSA content deceases the density of the composite. The Al6061–SiC composite density decreases by 1.11%, 2.68%, 4.39%, 5.87%, and 6.58% by increasing the CSA content from 2, 4, 6, 8 to 10 wt%, respectively.

Figure 12 shows the hardness values of Al6061 and its composites. It can be noted that with increment in wt% of CSA and SiC, the hybrid composite hardness increases. An increment in hardness by 37% (70 Hv) with an addition of 10% SiC was noticed. The increase in the CSA content from 2 to 8 wt% with an increment of 2 wt% in the Al6061–SiC matrix results in the increments of hardness by 31.5%, 36%, 39%, and 46%, respectively. Therefore, the hardness of the hybrid composite is improved by the SiC and CSA addition. However, the10% CSA addition decreases the hardness by 5%. The hardness of the Al6061–SiC–8% CSA hybrid composite is higher than that of the Al6061–SiC–10% CSA composite. The reason for the lower hardness with 10% CSA is the agglomerations of the reinforcement during casting.

Figure 13 depicts the Al6061 composite tensile strength. The tensile strength of the hybrid composites improved with the CSA addition. Compared to with the Al6061casting, the tensile strength increases by 18.26% for the Al6061–SiC composite. Further, with the addition of 2, 4, 6, and 8 wt% CSA, the hardness increases by 13.13%, 28.15%, 36.44%, and 47.31%, respectively, compared to that of the Al6061–SiC composite. Lakshmikanthan and Prabhustudied the mechanical and tribological behaviors of aluminium Al6061–CSA composite [128]. The uniform CSA particle dispersion in the matrix and the better interfacial bonding between the reinforcement particles and the matrix phase cause the hardness increment. Satheesh and Pugazhvadivu researched, for 10 wt% CSA content, a 14.8% decrement in tensile strength observed compared with that of the Al6061-SiC–8% CSA composite [125]. Agglomeration and void formation are responsible for this tensile strength reduction.

Figure 14 shows different weight percentages of reinforcements and composite elongation characteristics. It can be seen that with the increase of CSA content, the composite elongation decreases. The highest decrement (42.35%) in elongation is observed for the Al6061–SiC–10% CSA hybrid composite compared to that for Al6061. The percentage of elongation is an indicator of ductility. Kumar et al. investigated the characteristics of Al6061–WC–Gr hybrid MMCs and observed the same reduction in ductility [129].

Gopalakrishnan et al. determined specific strength and wear rate variations with TiC content [130]. Figure 15a,b shows AMMCs tensile strength (average) and ductility variations with different TiC percentages. It is observed that as the TiC percentage increases, the specific strength increases. This increment in strength is attributed to the resisting tendency shown by TiC. Figure 15b shows the addition of TiC reduces the % elongation.

Hybrid reinforcement offers high flexibility. Hybrid effectiveness generated by combining reinforcements in hybrid composites result in excellent mechanical properties as compared to that obtained in single reinforcement composites.

Premnath et al. fabricated hybrid MMCs using various weight fractions of Al_2_O_3_ (5%, 10%, and 15%) and a constant weight fraction of graphite (5%) using stir casting [103]. The hybrid combination of graphite and Al_2_O_3_ in AMMCs as a reinforcement results in attractive tribological and mechanical properties. The hardness and the density increase with the increase in the Al_2_O_3_ content. The uniform dispersion of particles and the better interfacial bonding improve the mechanical properties of the composites. The increase in the hardness with the increase in the AL_2_O_3_ content, because for the dislocation flow in the aluminium matrix, Al_2_O_3_ acts as a barrier. Due to the higher density of Al_2_O_3_ particles, the density of the composites increases.

Santosh et al. enhanced the mechanical properties with the addition of SiC and graphite in A6063 [131]. Al6063 reinforced with SiC (2%) and graphite (1, 2, and 3%) were fabricated by stir casting. The result showed SiC particles strengthen the composites, but whole mechanical property is decreased by 3% of graphite. Without the graphite tensile strength increase by 57.5% but for 3% of graphite added, the tensile strength decreases. The CS increases by 21.4% with 2% of SiC and 2% of graphite, but the CS decreases with 3% of graphite. The flexural strength increases by 125.3% with 2 wt% graphite but decreases with 3 wt% of graphite. The hardness increases from 42 to 55.9 for 0 to 2 wt% of graphite. With the increase in graphite content, the mechanical properties decrease due to the increase in porosity, as the graphite content increases.

The hybrid composite superior properties enable them to be used for a variety of applications. Guan et al. successfully produced Al6061 Al/5 vol% ABOw and 15 vol% of SiC particle hybrid composites by the semi-solid stirring method with 10–30 µm in length, 5 µm in diameter of ABOw, and 8–14 µm in range of SiC particles [132]. The fabricated hybrid composite tensile strengths were enhanced at different stirring temperature (630 to 680 degree centigrade) for 20 to 30 min stirring time. By increasing the stirring temperature from 630 to 680 °C, the UTS increases by 57.5%, 16.7%, and 28.5%.

Liu et al. critically analyzed the microstructural and mechanical properties of B_4_C and MOS_2_-reinforced AMMC hybrid composites by stir casting routes [111]. A7075 aluminium alloy with B_4_C as the reinforcement and MoS_2_ as the lubricant at various weight percentages of 4%, 8%, and 12% was investigated. The authors found that the reinforcement particles are uniformly dispersed as fine dendrites in the matrix. In comparison to monolithic alloy, the reinforced composites’ hardness, compressive, and tensile strength hall are improved by adding the reinforcement. The addition of a solid lubricant (MoS_2_) to the matrix alloy, as well as hard ceramic reinforcement particles (B_4_C), improves the wear resistance and the coefficient of friction of aluminium hybrid composites significantly. The load transfer mechanism between the matrix and the reinforcement particles improves the interfacial bonding, grain size, and the strain gradient, strengthening the contributions of the composites to the enhancement of the yield and tensile strength of the composites.

Many researches find that the addition of nanosized reinforcement enhances mechanical properties of composites. Girisha et al. fabricated a multiwall carbon nanotube (MWCNT)/AMMC by stir casting routes and studied its mechanical property [133]. The addition of small MWCNTs in the matrix increases the hardness to great extent. In addition, with wt% of MWCNTs increases, the yield strength increases.

Logesh et al. fabricated AlN/MWCNT/graphite/Al composites via stir casting route [134]. The reinforcements (AlN and MWCNT) were added as 0.5, 0.75, 1, and 2 vol%, and the graphite was maintained at 0.5 vol%. The authors found that, in the composites, the morphologies of the varied shaped reinforcements have a substantial impact on the mechanical characteristics, and better mechanical properties were reported. Integrated-shape reinforcements result in particle strengthening, while AlN results in grain refinement.

### 4.3. Tribological Studies of AMMCs

Since tribology is the science of interacting surfaces in a relative motion, it includes wear, friction, and lubrication. Aluminium alloys have long been utilized as a matrix material and have been reinforced with a variety of materials such as SiC, Al_2_O_3_, TiC, alumina, and B_4_C. Because of their excellent tribological qualities, these AMMCs are used in high-friction environments. The tribological properties of AMMCs are influenced by a variety of parameters, including applied load, sliding distance, environmental conditions, surface quality, geometry of reinforced particles, and reinforcement weight percentage. Sujan et al. fabricated AMMCs with SiC and Al_2_O_3_ and evaluated the wear rate of AMMCs with SiC as a reinforcement [25]. This study provides the following conclusions: (a) the wear rate increases, as the grinding speed (rpm) increases; (b) AMMCs with SiC as reinforcement particles have lower wear rates than AMMCs with Al_2_O_3_ reinforcement particles; and (c) the wear rates for both types of composite materials decrease significantly, as the particle reinforcements increase. For example, the wear rate of Al–15% SiC composites is only 0.126 mm/m, which is less than half of the wear rate of 3–1005 Al (0.255 mm/m).

Mondal et al. examined the influence of the load and the abrasive particle size on the wear characteristics of an Al alloy–Al_2_O_3_ particle composite [135]. The load application significantly influences the Al-alloy wear rate and is a highly influential parameter dictating the wear behavior. This study provides the following conclusions: (i) wear rate is dictated by load followed by abrasive size; (ii) compared to the wear in the alloy, the impacts of the abrasive size and the load on wear of composite are more prominent; (iii) for a certain combination of abrasive size and load, the wear rate of the composite may be higher than that of the alloy. 

Kumar et al. analyzed the mechanical and dry sliding wear of Al6061–SiC composites [136]. The hardness and the UTS of the Al6061–SiC composites were increased with the increasing SiC content. The wear property of the composites was superior. Figure 16 shows volumetric wear loss vs. load. The load application significantly influences the Al-alloy wear rate and is a highly influential parameter dictating the wear behavior. Figure 17 depicts the wear rate variation of the matrix and its composites with the SiC reinforcement. 

The increment in SiC volume fraction causes a decrement in volumetric wear loss of composites. However, compared to A6061 alloy, the composites have a lower volumetric wear loss for a given reinforcement content. With the increment in SiC, the hardness increases which increases the wear resistance and hence the volumetric wear loss decreases. Tyagi et al. investigated the tribological characterization of Al–TiC composites based composites [137]. With a normal load, the wear rate varies, which is in conformity with Archard law. In addition, in comparison to the alloy, the composites have a lower wear rate. The higher the load, the higher the wear (volumetric loss) for the matrix and its composites. However, the composite wear resistance is much higher compared to the matrix alloy at all loads. Alpas and Zhang reported that the microstructure (particle size and volume percentage) and counterface material have an impact on the sliding wear resistance of particulate-reinforced aluminium matrix composites [138]. Three different wear regimes were identified under different applied loads. At a lower applied force, reinforcing particles bear this applied load, such that the AMMCs wear resistance is higher than that of the Al alloy.

Ceschini et al. investigated the friction and wear behavior of C4 Al_2_O_3_/AI composites under dry sliding conditions [139]. Few researchers reported that wear resistance can be increased to 70% by reinforcing the ceramic phase. Uyyuru et al. researched the tribological behavior of an Al-composite/brake pad tribocouple is affected by the reinforcing volume fraction and size distribution [140]. The particle volume fraction results in an increment of dry sliding wear resistance. The higher the volume fraction, the higher the friction coefficient.

The worn out surfaces of A6061 and SiC-reinforced composites at a force of 60 N with a 6 km sliding distance are depicted in Figure 18 [136]. The existence of wear debris particles signifies adhesive wear and abrasive wear with asperities plastic shearing during wear tests. The reinforcements of SiC and Al_2_O_3_ behave as a secondary abrasive against the counter face that results in an increment of the counter face wear. In addition, the role of tertiary abrasive to the matrix and the reinforcement is played by reinforcement detached as wear debris. At a low load (60 N), 6 wt% SiC composites show a small wear loss. At a high force, the worn surface of the composites containing a lower volume fraction of SiC in the matrix alloy has a higher tendency to form grooves, which undergo plastic deformation resulting in severe wear. The SEM images shown in Figure 18a–d illustrate this phenomenon. With the increase in SiC content, the grooving tendency in the composite worn surface is reduced, signifying a lower material removal rate compared to that in the material matrix having no reinforcement as indicated in the SEM image.

Satheesh and Pugazhvadivu investigated the wear loss variation with different loads [126]. Figure 19 shows wear loss variations under 10 and 20 N loads. An increment ofwt% CSA results in a decrement of wear in the hybrid composites. The hybrid composites wear loss declines with the increase in wt% CSA. It was noticed that the soft particles in CSA come in between mating surfaces and thereby decreases the hybrid composites wear loss, due to the minimization of the friction. The increase in applied load increases the wear loss due to the increase in the friction between the pin surface and the disc. The wear rate variation with wt% CSA under 10 and 20 N loads (shown below) indicates that the addition of CSA decreases the wear rate but the wear rates under loads of 10 and 20 N for 8 and 10 wt% CSA is almost equal, indicating that up to 8 wt% of CSA results in a significant decrease in wear rate. The addition of SiC and CSA particles enhances the wear resistance, which is indicated by the decrease in the plastic flow of the material running parallel to the sliding direction.

Radhika et al. examined the tribological behavior of an aluminium alloy (Al–Si10–Mg) strengthened with alumina and graphite [141]. To determine the effect of factors on the wear rate and the coefficient of friction, they employed an L27 array with Taguchi’s technique. According to the authors, the most important influencing element on wear rate is the sliding distance (46.8%), followed by the applied load (31.5%) and the sliding speed (14.1%), and for the coefficient of friction, the sliding distance contributes 50%, the applied load contributes 35.7%, and the sliding speed contributes 7.3%. They also discovered that graphite has a considerable impact on the wear resistance.

Das et al. studied the abrasive wear characteristics of an aluminum alloy-based composite reinforced with alumina and zircon sandseparately [55]. The wear resistances of both composites increase, as the particle size of the reinforcement decrease, according to the authors. The authors discovered that the bonding among aluminium and zirconisis better than that among alumina particles, resulting in a zircon-reinforced composite with better wear characteristics than an alumina particles-reinforced composite.

Kenneth et al. [142] compared to single reinforced AMMCs with a reinforcement of a mixture of agro waste derivatives and synthetic ceramic material to impart better wear characteristics to the composites. The hybrid reinforcement shaving an enhanced wear resistance show a higher abrasive wear compared toa adhesive wear because of little debris on the composite surface. The friction coefficient is increased due to the debris adhesion on the composite surface, resulting in a high wear rate. Figure 20a–d depicts a hybrid Al–Mg–Si/bamboo leaf ash (BLA)–Al_2_O_3_ worn-out surface. The hybrid composites with 2 and 3 wt% BLA depict fewer debris adhered to the surface compared to those with 4% BLA hybrid composites.

Raju et al. studied the tribological behavior of an Al-1100–CSA composite at elevated temperatures [143]. At an elevated temperature for all applied pressures, the Al(5%, 10%, and 15%)–CSA composites have shown better wear resistances. Higher volume fractions of the reinforcement in Al-1100 matrix composites have acquired greater wear resistances at all applied pressures when heated to 150 °C. The wear/wear rate for Al–15% CSA is found to be the lowest. At all the pressures, the composites containing higher percentages of reinforcement (10% and 15% CSA) have demonstrated to have an increased wear resistance.

Recent investigations have revealed that a variety of natural fiber products can be used in advanced engineering applications. Coyal et al. fabricated cost-effective AMMCs with SiC and jute ash particles with enhanced properties via stir casting routes [144]. The wear behavior was investigated using a pin-on-disc tribometer, and it was discovered that adding reinforcement particles increases wear resistance. When compared to the base material, the wear resistance of the fabricated composite samples is about four times higher. The wear resistance of the matrix metal enhanced with SiC is the best of all produced samples. Due to the establishment of a mechanically mixed layer, the coefficient of friction drops dramatically in the presence of the reinforcement. Mahesh et al. manufactured a jute rubber composite with various stacking sequences and investigated the impact of process parameters on the composite wear behavior [145]. In comparison to load and composite configuration, they found that the abrading distance had the greatest impact on the wear rate. Kumar et al. employed an Al–4.5% Cu alloy that was stir-cast and reinforced with 2%, 4%, and 6% BLA [146].

After reviewing the literature on AMMCs with various reinforcing particulates through stir casting, it was discovered that the bulk of studies tried to improve their mechanical and tribological properties due to the addition of reinforcement particles. Despite the widespread use of SiC, Al_2_O_3_, B_4_C, B_4_C, and TiC as reinforcements in AMMCs, researchers are increasingly more interested in the combinations of various reinforcements such as hybrid composites/nanocomposites to obtain enhance properties via stir casting methods. Nowadays, many researchers focus on natural fiber as reinforcement particles for AMMCs. It also takes into account the use of industrial and agricultural leftovers as reinforcements. This demonstrates the growing tendency of using both organic and inorganic elements as reinforcing components in AMMCs.

## 5. Conclusions and Future Work

The goal of this review is to look at the fundamental microstructural, mechanical, and tribological behaviors of AMMCs with various reinforcements via the stir casting process. The literature gives an overview and the future importance of single, dual, and hybrid AMMCs, shown as follows.
Dual- and hybrid-reinforcement metal matrices having superior properties as compared to single-reinforced aluminium matrix composites.Hybrid AMMCs with various reinforcement particles result in a more homogeneous microstructure into the matrix and can be fabricated through the stir casting technique.The processing parameters of stir casting, such as reinforcement size, stirrer speed, stirrer blade design, stirring time, and melt temperature, have significant impacts on the properties of AMMCs.The distribution of the reinforcement into the matrix brings an enhancement in mechanical properties such as microhardness, tensile, and ductility.The tensile strength and the ductility of composites increase integrally, as the grain refinement enhances the plastic deformation of the composites.The wear resistance of hybrid composites improves due to the strong bonding between the reinforcement and the base metal matrix.The low-cost leftovers of agricultural and industrial waste use in AMMCs as reinforcements improve the mechanical and tribological properties.

As per literature review, numerous reinforcements in different volumes and weight percentage make AMMCs widely applicable. Future work is need to be carried out on the combination of soft and hard reinforcements for enhancing the microstructural, mechanical, and tribological properties of A6061 alloy, which is widely applicable in aerospace, marine automobile, and structural areas.

## Figures and Tables

**Figure 1 materials-14-06386-f001:**
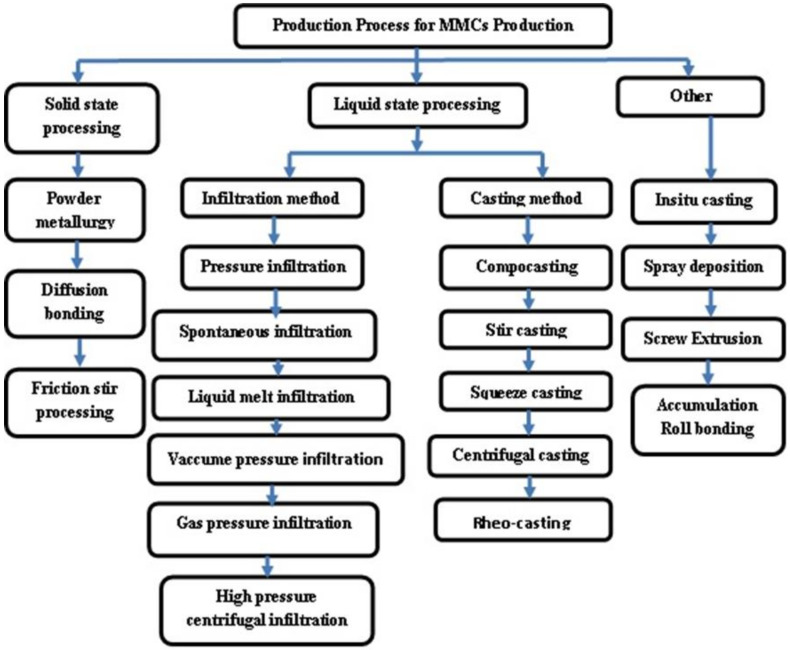
Flow chart of various production processesfor AMMCs [16].

**Figure 2 materials-14-06386-f002:**
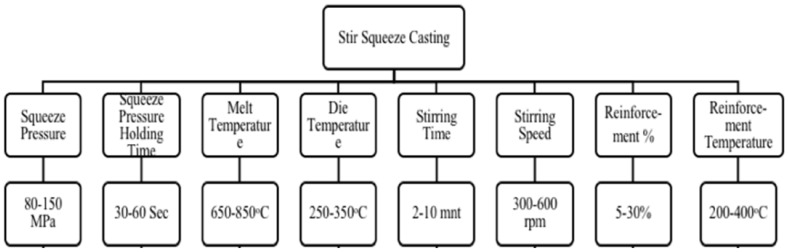
Important process parameters of stir squeeze casting [16,27,62,63,64,65,66,67,68].

**Figure 3 materials-14-06386-f003:**
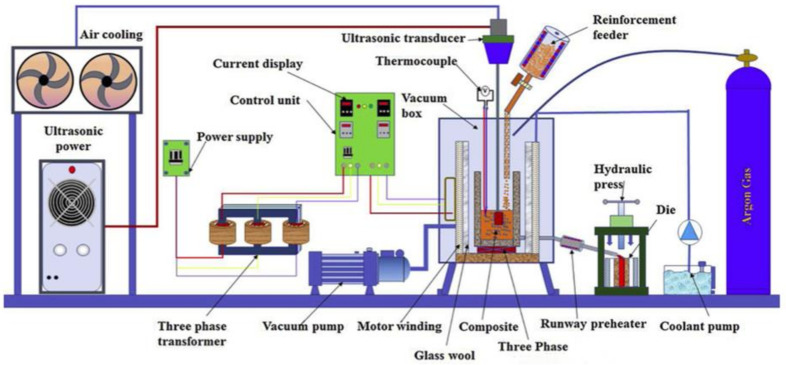
Stir casting furnace design [16].

**Figure 4 materials-14-06386-f004:**
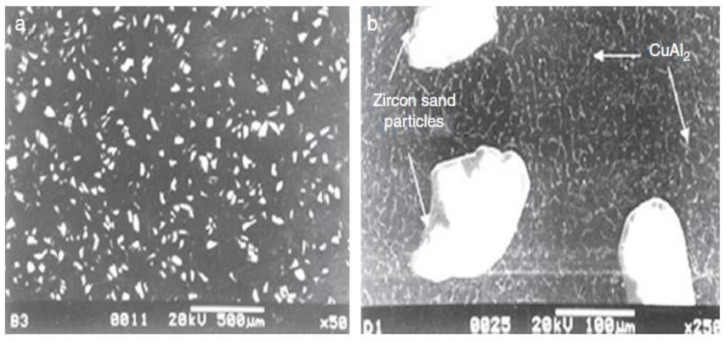
SEM images of MMCs showing (**a**) the identical distribution of zircon particles and (**b**) the existence of intermetallic segment [116].

**Figure 5 materials-14-06386-f005:**
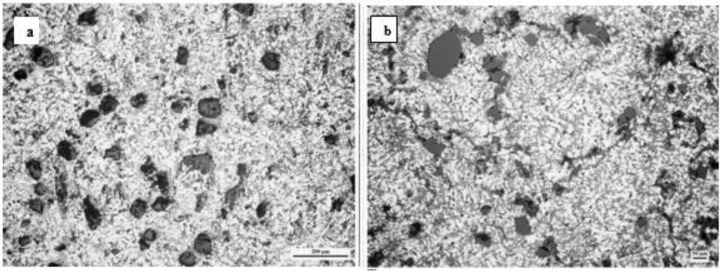
(**a**) Optical micrograph of DPS1 composites having 12% coarse and 3% fine particles showing eutectic Si morphology changes to globular from acicular in neighborhood to particles; (**b**) optical micrograph of DPS2 composites having 12% fine and 3% coarse particles displaying fine particles clustering [55].

**Figure 6 materials-14-06386-f006:**
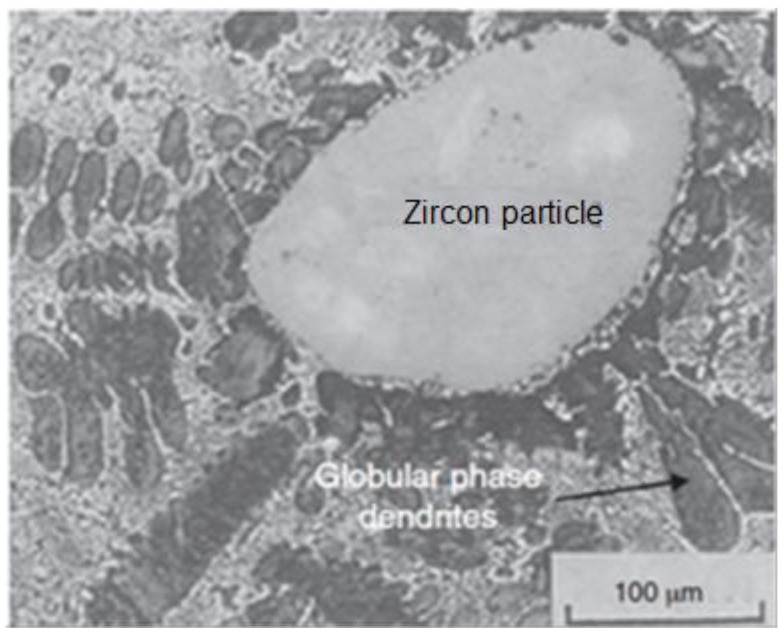
Optical micrograph of ZAS alloy showing a hypoeutectic Zn-rich primary-phase dendrites distribution [119].

**Figure 7 materials-14-06386-f007:**
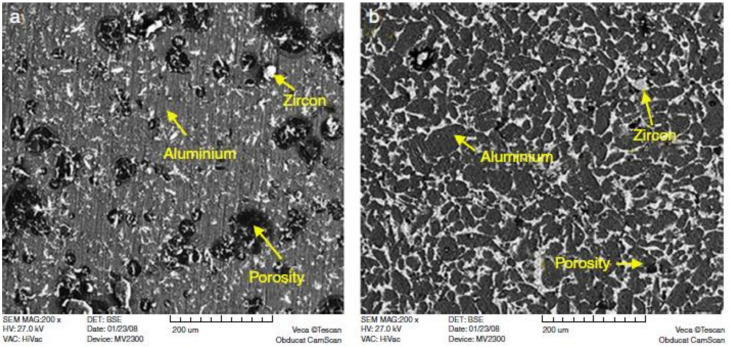
SEM images of the composites (sintered at 600 °C) (**a**) having 5 vol% zircon particles and (**b**) 15 vol% zircon particles [123].

**Figure 8 materials-14-06386-f008:**
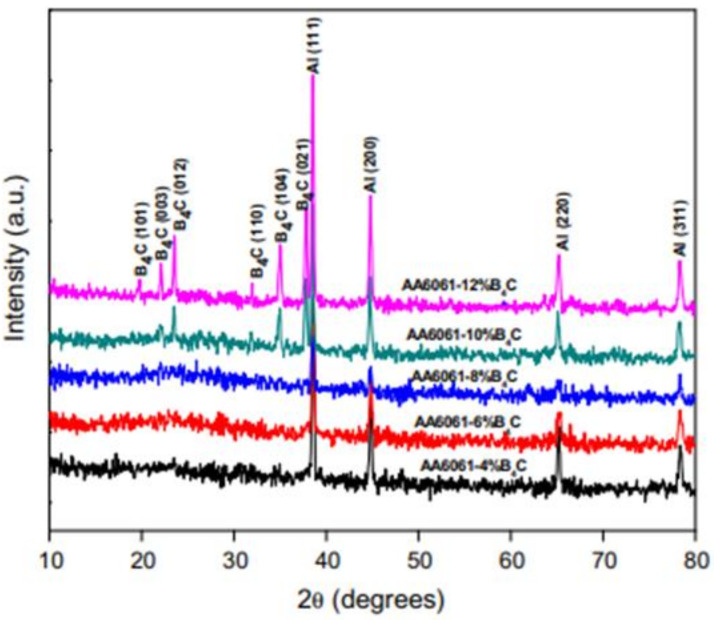
XRD patterns of AA6061–B_4_C composites [124].

**Figure 9 materials-14-06386-f009:**
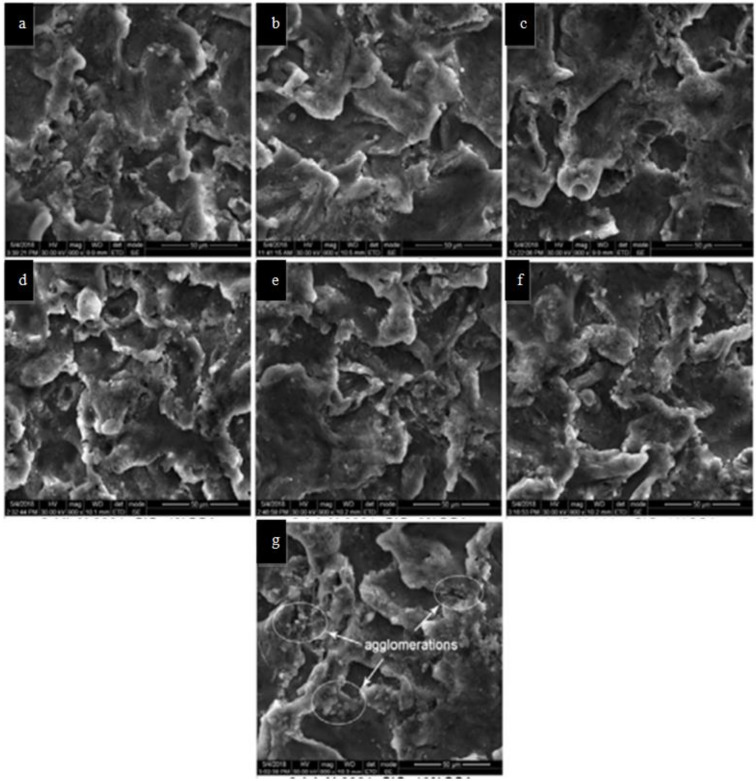
SEM micrographs of Al6061 and its composites (**a**) Al6061 (**b**) Al6061+SiC (**c**) Al6061+SiC+2% CSA (**d**) Al6061+SiC+4% CSA (**e**) Al6061+SiC+6% CSA (**f**) Al6061 +SiC+8% CSA (**g**) Al6061+SiC+10% CSA [126].

**Figure 10 materials-14-06386-f010:**
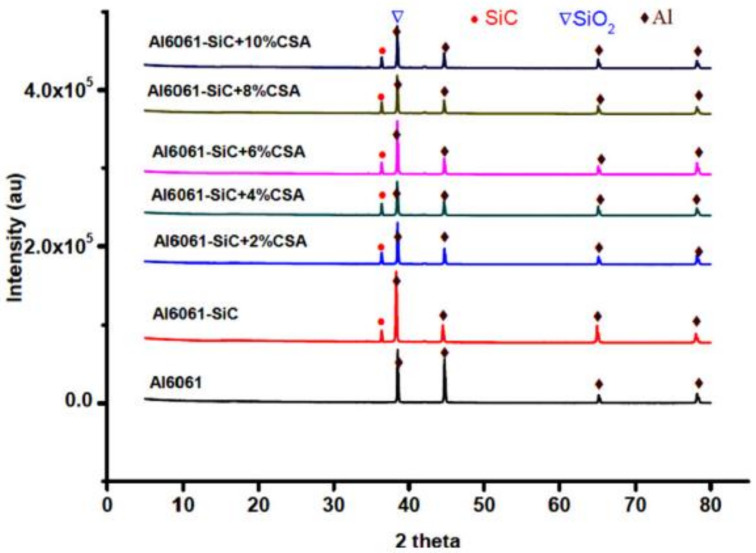
XRD peaks of Al6061 and its composites [125].

**Figure 11 materials-14-06386-f011:**
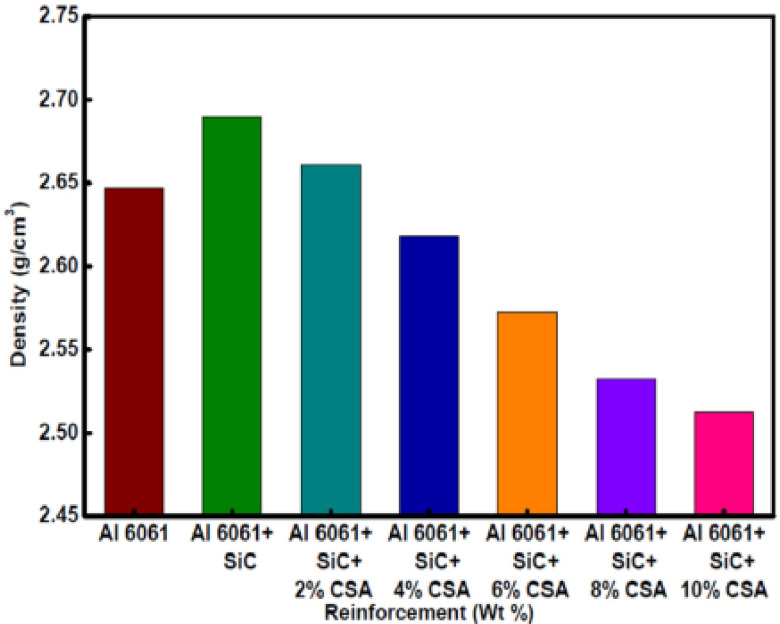
Densities of Al6061 and its composites [125].

**Figure 12 materials-14-06386-f012:**
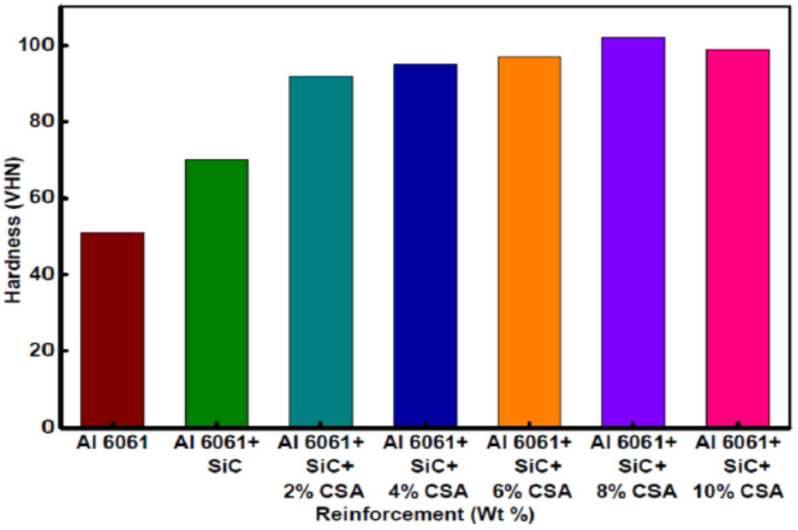
Hardness values of Al6061 and its composites [125].

**Figure 13 materials-14-06386-f013:**
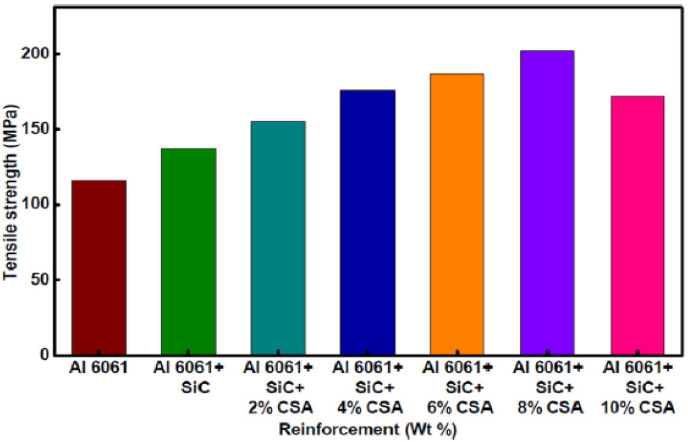
Tensile strengths of Al6061 alloy and its composites [125].

**Figure 14 materials-14-06386-f014:**
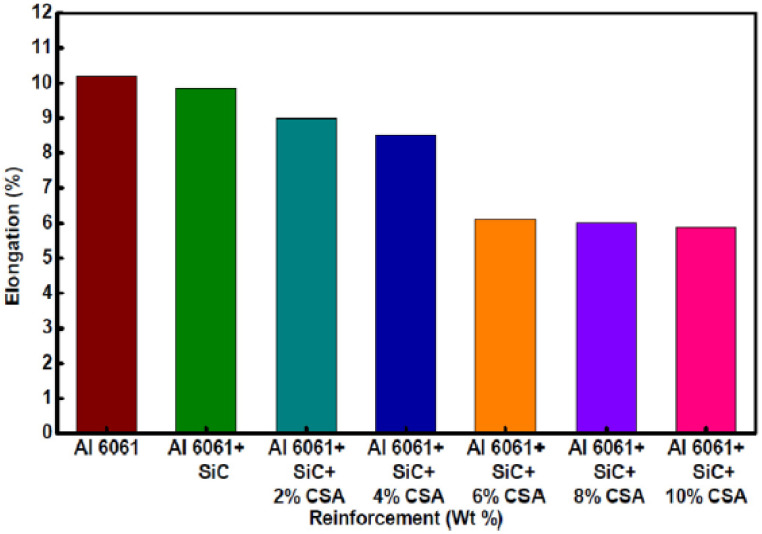
Elongation percentages of Al6061 alloy and its composites [125].

**Figure 15 materials-14-06386-f015:**
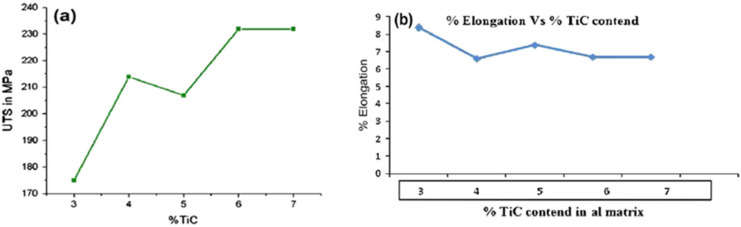
Effects of TiC reinforcement on the mechanical properties of AMMCs: (**a**) tensile strength; (**b**) ductility [130].

**Figure 16 materials-14-06386-f016:**
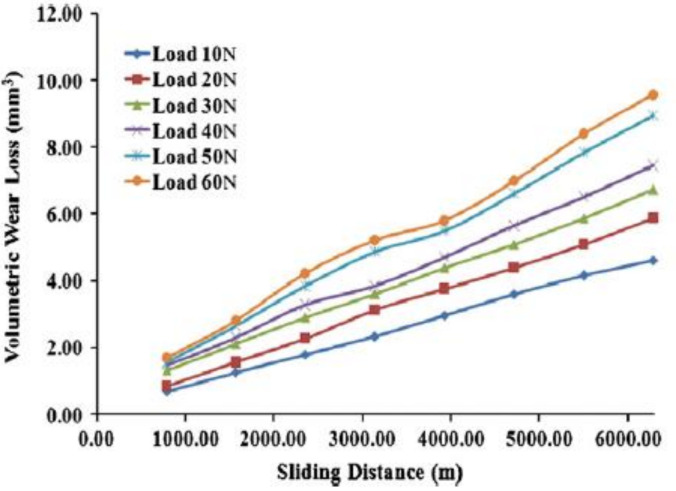
Load application effect on the wear (volumetric loss) with the increment in the sliding distance on Al6061–6 wt% silicon carbide-reinforced composites [136].

**Figure 17 materials-14-06386-f017:**
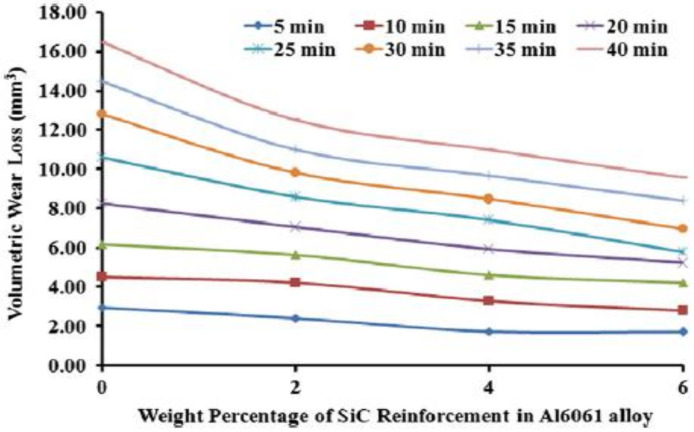
The wear (volumetric loss) of cast Al6061 alloy with increase in % of SiC-reinforced composites [136].

**Figure 18 materials-14-06386-f018:**
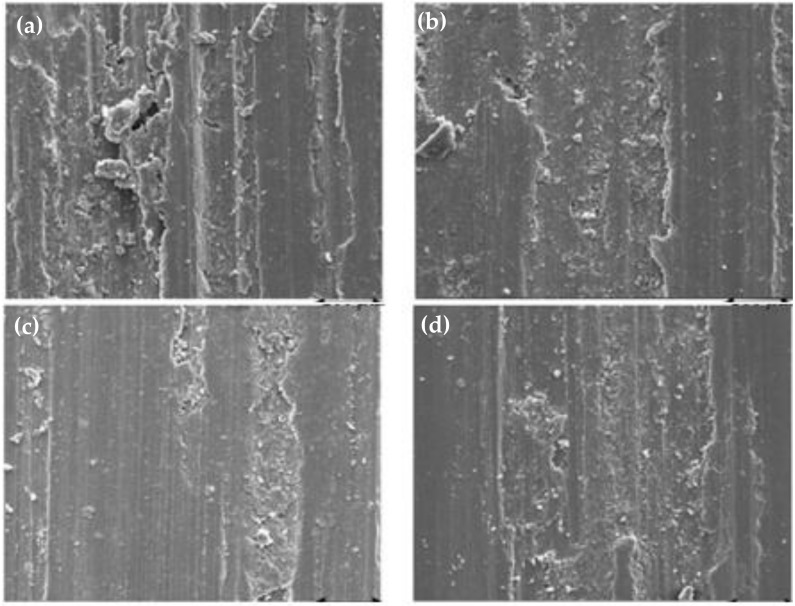
SEM images of the worn surfaces of Al6061 alloy and Al6061/silicon carbide-reinforced composites at a 60 N applied force and a sliding distance of 6 km (**a**) Cast Al6061 alloy (**b**) Al6061-2 wt% SiC (**c**) Al6061-4 wt% SiC (**d**) Al6061-6 wt% SiC [136].

**Figure 19 materials-14-06386-f019:**
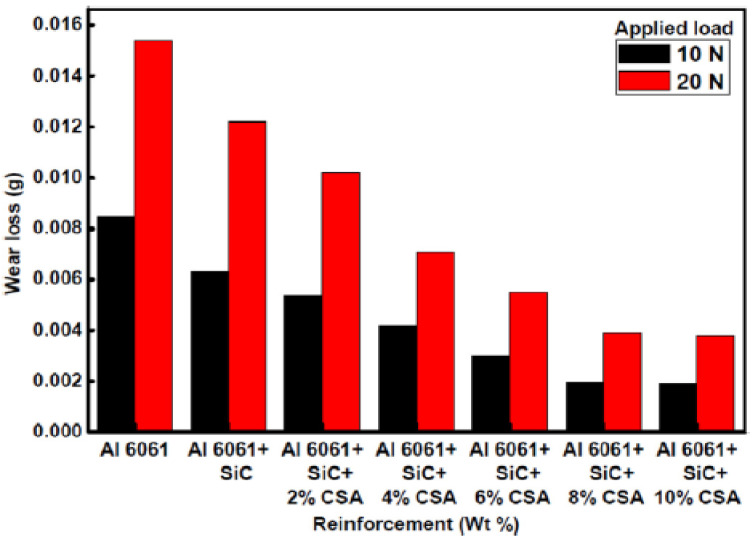
Wear losses of Al6061 and its composites [125].

**Figure 20 materials-14-06386-f020:**
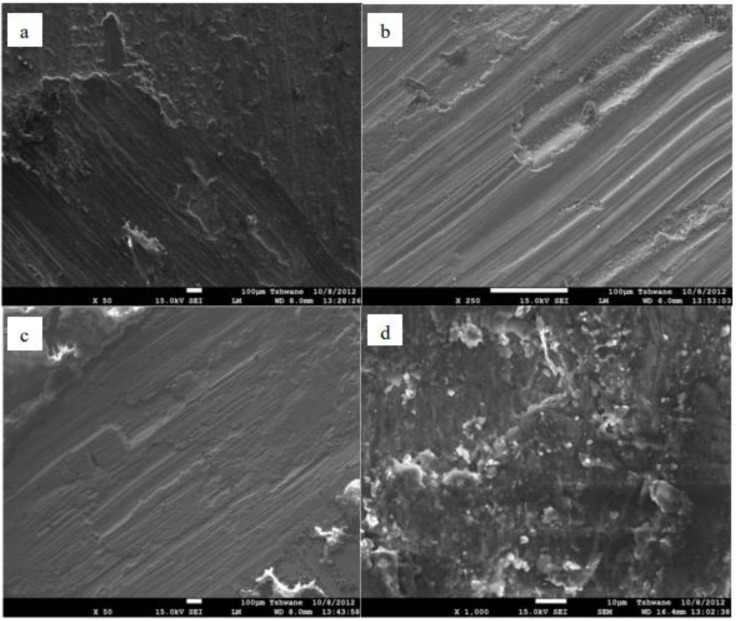
SEM micrograph of worn-out surfaces of (**a**) the single reinforced aluminium–Mg–Si/10 wt% Al_2_O_3_ composites, (**b**) hybrid reinforced Al–Mg–Si/2 wt% BLA (bamboo leaf ash)-8 wt% Al_2_O_3_ composites, (**c**) hybrid reinforced Al–Mg–Si/3 wt% BLA-7 wt% Al_2_O_3_ composites, and (**d**) hybrid reinforced Al–Mg–Si/4 wt% BLA–6 wt% Al_2_O_3_ composites [142].

**Table 1 materials-14-06386-t001:** Various fabrication techniques with unique features and costs [60,61].

Method	Cost	Application	Remarks
Stir casting	Least cost	Applicable for mass production	Appropriate for particulate reinforcement in aluminium metal matrix composites (AMMCs), depending on matrix metals, reinforcement properties, and casting speed and duration
Powder metallurgy	Medium cost	Used in the production of parts that are heat-resistant (e.g., piston and valves)	Matrix metal and reinforcements are required in powder form. Due to the absence of a reactive zone, the manufactured metal matrix composite (MMC) has a high strength.
Diffusion bonding	High cost	For sheets, vane shafts, and blades, this process is used.	Suitable for Matrix Metal in the form of sheets; Reinforcements in the form filaments.
Liquid infiltration	Low cost	Structural component manufacturing such as rods, beam, and tubes	A reinforcement in form of filament is used.
Squeeze casting	Medium cost	Engine parts production, e.g., piston, connecting rods, and cylinder head	Generally preferred for the type of reinforcement. Used for mass-scale production.
Compocasting	Low cost	Applicable to the automobile industry	Conducive for discontinuous fibre in particle form
Insitu	Medium cost	Applicable to the automobile sector	It results in a homogeneousdistribution of reinforcing particles.
Ultrasonic-assisted casting	High cost	Suitable for mesh-shaped components. Preferred for large-scale manufacturing.	Almost a uniform dispersion of particulate reinforcements.

**Table 2 materials-14-06386-t002:** AMMCs fabrication with different reinforcements through stir processing routes.

Types of Composites	Method	Outcomes	References
Al6061/Al_2_O_3_ (45 µm)/Gr (60 µm)/Gr (60 µm)	Stir casting	Increased hardness increases/increased density/good wear behavior	[103]
AlSi_18_CuNi/Al_2_O_3p_	Stir casting	Two wt% of Al_2_O_3_ increases the tensile strength (505 MPa) and hardness (123 Hv) as compared to the unreinforced matrix.	[104]
Al6061/nano-RHA	Ultrasonic and stirring	Increased microhardness	[105]
Al6061/B_4_C (10 µm)	Stir casting-modified method	Enhanced macro- and microhardness values and enhanced tensile strength	[106]
Al6061/B_4_Cp (88 µm) (5 and 7 wt% of B_4_C)	Two-Stage Melt Stirring	B_4_C improves the compressive and tensile strength. Two-stage melt stirring homogeneously distributes B_4_Cparticles without agglomeration or clustering.	[106]
A2219/15 vol% silicon carbide particle (SiCp)/3 vol% Gr (25 µm)	Liquid stir casting technique	Less flank wear is observed while machining the composite containing graphite as one of the reinforcement. Compared to the carbide tool, coated carbide showed less flank wear. PCD tools are used for mass production.	[107]
A356/5 wt% B_4_C (20–50 mm)/20 wt% A356 (32–80 µm)	Semi-solid material (SSM) stirring technique	A higher stirring speed, a higher stirring time, and a lower stirring temperature promote the uniform distribution of the reinforcement. The pretreatment of SiC improves the wettability between Al356 alloy and SiC. The addition of Mg improves the interfacial bonding and hence improves the hardness elongation and the tensile strength.	[108,109]
AM60/Al_2_O_3_p (25 nm) and Al2024/Al_2_O_3_ (16, 32, and 66 µm)	Stircasting method	Due to the more effective grain refinement in AM60 MMCs than in cast AM60, the MMC (AM60 and 1 wt% Al_2_O_3_) has a higher ultimate tensile strength (UTS), a higher Yield Strength and a higher ductility value (p107%, p135%, and p245%, respectively). The tensile strength increases with the decreasing particle size and rising weight proportion of particles.	[36,110]
Al7075/B_4_C/MoS_2_	Stir casting	The mechanical and physical properties of the reinforced composites are enhanced by adding the reinforcement. Enhancement in wear resistance is noticed.	[111]
Al6061/seasand/SiC/Al_2_O_3_	Stir casting	The Al6061/sea sand composite showsa higher density a and lower porosity as compared to the Al6061/Al_2_O_3_ and Al6061/SiC composites.	[112]
Al2124/Al5083 and Al6063 reinforced by SiC (157–511 µm)	Stir-casting method	The impact strength increases with particle aggregation, while it isreduced with an increased extrusion ratio and an increased particle size. In addition, the matrix-reinforcement bonding and particle cracking have effects on the impact characteristics of the composites.	[22]
Al/Al_2_O_3_/SiC	Stir casting	Wear resistance increases with the increase of the reinforcement addition. The Al–SiC–Al_2_O_3_ composite is a green replacement for grey cast iron which can be used in the manufacturing of car disc brake rotors.	[25]
Al384/SiC (60 µm) and A356/SiCp (4 µm)	Stir casting	The stirring time and speed have a great impact on the microstructure and hardness. Smaller stirring times and stirring speeds result in particles clustering, whereas higher stirring times and stirring speeds result in a uniform distribution of the reinforcement. A good combination of the increased tensile elongation and the UTS is achived through the Rheo stir casting process.	[27,28]
Al/SiC	Stir casting	The surface finish is greatly influenced by the cutting speed and the percentage of the reinforcement followed by the feed rate.	[29]
Al356/SiCp/B_4_Cp	Stir casting	The surface finish is least influenced by the depth of the cut and is highly influenced by the cutting speed.	[42]
Al6063/ZrSiO_4_ and Al_2_O_3_	Stir casting	Hybrid composites outperformed zircon sand and alumina particles-reinforced composites. The hardness and the tensile strength of the compositewith the reinforcement wt% combination of 4% ZrSiO_4_ + 4% Al_2_O_3_ are higher.	[54]
Al–7Si–0.35Mg/fly ash	Liquid-metal stir casting, compocasting and modified compocasting	In liquid-metal stir-cast composites, interfacial interactions at the fly ash particle–matrix intefacelead to the synthesis of MgAl_2_O_4_. Well-dispersed, reasonable agglomeration and porous-free composites areformed.	[57]
Al/15 wt% flyash	Stir casting	The wear resistance of the flyash-reinforced material improves as the flyash content increases, but decreased when the load and the track speed increase. The base alloy wears predominates as a result of microcutting, according to the microscopic analysis of worn surfaces, wear debris, and the subsurface. However, delamination, microcutting, oxidation, and thermal softening cause the MMCs to wear out. Corrosion increases as the amount of flyash has increases.	[59]

## Data Availability

All data are available within the manuscript.

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
