# Peer review of "A Contemporary Review of Aluminium MMC Developed through Stir-Casting Route"

_materials, 2021, doi:10.3390/ma14216386_

Round 1
Reviewer 1 Report
This review article lacks consistency, scientific approach and comprehensiveness, for example, the following comments if considered might improve the article.
- The sound of writing in both abstract and conclusions, looks like the authors are doing the experimental work of this study for example:
Abstract:
The various microstructural, tribological and mechanical properties were tested by suitable tester with consideration of different input parameter. (The statement indicate that you have tested)
The abrasive and erosive wear of differently fabricated aluminium matrix nano composite was analysed with tribo tester. (The statement indicate that you have analysed with tribo tester)
The reinforcement’s homogeneity fractural behaviour and worn-out surface mor-22 phology of test specimens was investigated through various characterization techniques and corre-23 lated with the obtained results. (The statement indicate that you have investigated)
Conclusions:
The A6061 reinforced metal matrix composite successfully fabricated through stir casting technique with evenly homogeneous distribution of reinforcement 1009 particles in matrix. (The statement indicate that you have fabricated)
- There is no caption or number for figures presented in page 2 (flow chart, it seems like this referred as Figure 1 in the text may be, not clear), page six two Figures without caption also or even number of figures.
- The cited reference brackets [] should come before the end of the statement not after throughout the manuscript like that []. Not like that. []
- The first citation for Fig. 1 it seems incorrected. “Poor interfacial reactions and wettability between matrix and reinforcement occur in MMC produced by ex-situ process while no such limitation found in in-situ as shown in Fig. 1.” Regardless of not having figure 1, there is no figure to show what mentioned in this statement.
- The resolution of the flow chart needs to be improved and to give number and caption for the figure and cite in the text.
- “important process parameters and additives of stir casting method are 206 discussed as shown in 2 and 3 [13]. Low strength property of pure Al inhibits its use 207 in commercial application.” Where is these figures and how the parameters are discussed in the figure?
- Section 2. Reinforcing materials in AMCs. Intended to give information about the different types of reinforcing material themselves however, the subtitles are mainly talking about the AMCs reinforced with these materials. Needs correction.
- All manuscript sections consist of truncated statements that need to be connected and properly discussed.
- Also all sections need to be introduced properly before stating the reported work in the topic for example :
Section 2.1 : started “Machinability and mechanical properties of silicon carbide (SiC) reinforced AMMC 65 were investigated [25].”
Section 2.2 : started “Investigator reported higher Al2O3 percentage decreases the inter particle spacing be-110 tween nucleated micro voids resulting in decreased fracture toughness of MMCs [41].”
Section 2.2 : started “Author studied factor affecting tri-modal aluminium metal matrix composites 136 strength and reported that high strength is obtained with amorphous and crystalline aluminium nitride and Al4C3, [57]”
- Please unify the abbreviation throughout the manuscript “AMMCs”
- Table 2 is too long and not well presented. The table need to be reduced and the information given about the property improvement can be discussed in the text.
- Although the article is mainly concerned with AA6061 MMCs, some sections are presented in a general way related to all types of AMMCs.
- Figure 9, 11, mentioned that OM and SEM images are presented while only OM micrographs are presented.
- Section “4. Microstructural characteristics of reinforced metal matrix composites” with this title only microstructural characteristics are expected to be discussed, however many mechanical properties subtitles are given, 1. Microhardness and density, 4.2. Tensile strength, 4.3. Fracture analysis, 4.4. Wear analysis
- Also the same for section 5. Properties of Aluminium metal matrix hybrid composites, many unrelated subtitles are discussed
- Figure 26, 27, 28 are of very bad resolution and incorrected reference citation ref
- All sections need improvement in the presentation, discussion and quality of writing.
Author Response
Thank you and the reviewer for a careful reading of the manuscript ID materials-1384812, and for the comments and suggestions who helped us increase the clarity of the presentation.
All the points raised by the reviewer has been addressed. In the revised manuscript these changes are highlighted in yellow. The reviewer comments and our detailed responses and explanations of the changes are given below attachment.

Reviewer 2 Report
The research article “A Contemporary review of Al6061 MMC developed through Stir-Casting Route“ is devoted to an aluminum alloy composites and methods of their production. Aluminum alloys and composites are a promising material for industry with excellent technological and mechanical properties. The authors have done a great job on the literature data analysis and structurization. I believe this work will contribute significantly to the development of this scientific area.
Below are drawn the notes and questions that should be clarified for the publication. My recommendation is to accept the article for publication after Minor Revision.
Line 46-47: I could not find the captions of figure 1. Add please.
Line 179-190: The authors write about the chemical compound "Zircon". Could you clarify what kind of material is it? Also, in this paragraph there are such terms as "alumina enhances" and “alumina reinforced composite”. Maybe authors mean Zirconium and Aluminum?
Line 218-220: I could not find the captions of figure 2 and 3. Add please. The design of the article is very inattentive.
Line 256: The data shown in table 1 belong to some articles? Please add references.
Line 365: On Figure 6, there is a typo in the inscription of the “Zircon particle”.
Figures 10, 15, 16, 17, 18, 19, 21, 22 have no confidence intervals for the experiment values. I understand that this is literary data, but it is not acceptable to use it. If authors use these graphs in review they should critically pay attention at this point.
On Figure 36c there is no dimension mark where 1 µm is indicated.
On Fig. 40, there are no reflections from the second phase in the electron diffraction patterns.
Author Response

(The authors gave the same response as above.)

Reviewer 3 Report
In the manuscript, the authors summarized the development of stir casting technique and reinforcement composites in the AMMCs. The effect of different composites and manufacturing processes on the AMMCs was discussed. In reviewing the manuscript, the following comments and suggestions are made.
- In the review version, the explanatory text is missed in the figure 1, 2, and 3, please double check.
- In figure 1, the “in situ casting, etc.” are categorized into “other techniques”, rather than “casting method”, why?
- In page 2 line 53, the abbreviation of “AMMCs” has been defined in the Abstract, please check. Also, in subtitle of Section 2, please define “AMCs”.
- It is recommended use the simple present to describe the findings of publications (i.e. not past tense).
- In page 3 line 81, how much IS and hardness increased?
- In section 2, the authors list the effect of several reinforcing materials on AMCs, and mainly focus on the description of the literature. It is recommended that authors can provide some depth discussion or profound summarization on these materials.
- The description of “Table 1” (page 7 line 254) orders after “Table 2” (page 5 line 217), please check.
- In section 4, characterization includes various fields, like structure, mechanical property, thermal property, chemical performance, etc. In this section, 4.1 to 4.4 are not the “microstructure characterization” techniques.
- In section 4, it is recommended to discuss the importance of “characterization”, in other word, build a relationship between the “microstructure” and “properties”.
- In section 5, also, it is recommended that authors can provide some depth summarization of aluminum metal matrix hybrid composites, rather than the microstructure and properties description.
- Please check carefully spelling and grammatical errors in the manuscript.
Author Response

(The authors gave the same response as above.)

Round 2
Reviewer 1 Report
It seems the article has been improved throughout and only very few corrections my be carried out before publication, for example:
- Figure caption no 1 correct "sir casting" to be Stir casting"
- The quality of Figure 10 must be improved.
- Details and scale bar in Figure 5, Figure 9 must be more clear and visible
- In Table 2, give a heading for each column and repeat on each page.
- In Figure 20, there is two scale bars in Figure 20b, which is correct.
Author Response
We are thankful to the reviewer for providing us with another chance to improve the manuscript. The authors are also grateful to the reviewer for the stupendous task of evaluating the manuscript and providing valuable and constructive inputs. The manuscript has been revised according to the reviewer’s comments/suggestions, and the corrections are highlighted in red colour.
